# Antioxidant, Anti-Inflammatory, and Anti-Cancer Properties of Amygdalin Extracted from Three Cassava Varieties Cultivated in Benin

**DOI:** 10.3390/molecules28114548

**Published:** 2023-06-05

**Authors:** Halfane Lehmane, Arnaud N. Kohonou, Atchadé Pascal Tchogou, Radiate Ba, Durand Dah-Nouvlessounon, Oscar Didagbé, Haziz Sina, Maximin Senou, Adolphe Adjanohoun, Lamine Baba-Moussa

**Affiliations:** 1Laboratory of Biology and Molecular Typing in Microbiology, Department of Biochemistry and Cell Biology, Faculty of Sciences and Techniques, University of Abomey-Calavi, Cotonou 05 BP 1604, Benin; halfanel@gmail.com (H.L.); loveraphi@yahoo.fr (R.B.); dahdurand@gmail.com (D.D.-N.); oscar_didagbe@yahoo.com (O.D.); sina.haziz@gmail.com (H.S.); 2Laboratory of Research in Applied Biology, Polytechnic School of Abomey-Calavi, University of Abomey-Calavi, Cotonou 01 BP 2009, Benin; kohonouarnaud60@gmail.com; 3Experimental and Clinical Biology Laboratory, National School of Applied Biosciences and Biotechnologies, National University of Science, Technology, Engineering and Mathematics (UNSTIM), Dassa-Zoumé 01 BP 1471, Benin; tchopass2@gmail.com (A.P.T.); senouxim@yahoo.fr (M.S.); 4Institut National des Recherches Agricoles du Bénin, Cotonou 01 BP 884, Benin; adjanohouna@yahoo.fr

**Keywords:** amygdalin, cassava, phytochemistry, biological activities, Benin

## Abstract

Given that cancer is a disease that is rampant in the world and especially in Africa, where the population has enormous difficulty treating it, plants are a safer and less expensive alternative. Cassava is a plant species valued in Benin because of its numerous medicinal and nutritional virtues. This study evaluated the biological activities of amygdalin from the organs of three cassava varieties most commonly produced in Benin (BEN, RB, and MJ). HPLC analysis was used to quantify amygdalin in cassava organs and derivatives. Phytochemical screening was performed to determine secondary metabolite groups. DPPH and FRAP methods were used to assess antioxidant activity. Cytotoxicity of the extracts was tested on *Artemia salina* larvae. The anti-inflammatory activity was evaluated in vivo in an albino mouse paw edema model induced by 5% formalin. The anticancer activity was evaluated in vivo on Wistar rats rendered cancerous by 1,2-dimethylhydrazine (DMH) using 5-fluorouracil as a reference molecule. The results showed that the organs of all three-cassava varieties contained glycosides, flavonoids, saponosides, steroids, tannins, coumarins, and cyanogenic derivatives. Young stems and fresh cassava leaves had the highest amygdalin concentrations, with 11,142.99 µg 10 g^−1^ and 9251.14 µg 10 g^−1^, respectively. The *Agbeli* derivative was more concentrated in amygdalin, with a content of 401.56 µg 10 g^−1^ than the other derivatives. The antioxidant activity results showed that the amygdalin extracts were DPPH radical scavengers with IC_50_ values ranging from 0.18 mg mL^−1^ to 2.35 mg mL^−1^. The cytotoxicity test showed no toxicity of the extracts toward shrimp larvae. Administration of amygdalin extracts from the leaves of BEN and MJ varieties prevents inflammatory edema. The percentages of edema inhibition varied between 21.77% and 27.89%. These values are similar (*p* > 0.05) to those of acetylsalicylic acid (25.20%). Amygdalin extract of the BEN variety significantly (*p* < 0.0001) reduces edema. Both BEN extracts inhibited cancer induction with DMH. In preventive and curative treatments, rats fed with amygdalin extracts showed low anti-cancer activity under the effect of DMH and a significant difference in biochemical results. Thus, the organs of all three cassava varieties studied have secondary metabolites and good antioxidant activity. The leaves contain high levels of amygdalin and can be used as anti-inflammatory and anticancer agents.

## 1. Introduction

Cancer is now a significant public health problem, with over 18 million new cases and 9.6 million deaths in 2018 [1]. Cancer is among the leading causes of morbidity and mortality worldwide [2]. It is characterized by the excessive proliferation of abnormal cells, which can be lethal if not effectively treated. This mortality remains high despite recent advances in treatment in developed regions in recent years. The number of new cancer cases per year worldwide has increased from 14 million in 2012 to more than 10 million in 2020 [3], with a rate of more than one million new cases recorded in Africa [4]. Africa, once considered the preserve of high-income countries, is not exempt from cancer. In Africa, more than 700,000 deaths were recorded in 2020 [4], and for the 2030 projections, the estimated figures are, among others, 1.4 million new cases and 1 million deaths [2]. More than 95% of cancer patients in African countries are diagnosed at an advanced stage [5]. Delayed diagnosis for these patients is due to insufficient awareness and a lack of qualified centers with well-trained personnel [5]. This delay is also dominated by financial problems, the lack of motivation of elderly patients, and the absence of the concept and word “cancer” in several African languages [6]. In Benin, since 2013, the number of cancer patients has been growing, and 1500 cases are recorded yearly in Cotonou, with 55% of deaths [7].

Modern medicine has already developed various alternatives against the disease, such as chemotherapy, surgery, gene therapy, radiotherapy, immunotherapy, and others. However, conventional cancer therapies are associated with a lack of selectivity and severe side effects [7]. These chemotherapies are therefore considered risky drugs. Indeed, they combine multiple risks concerning the environment, the caregivers, the patients, and, more generally, all users in healthcare institutions, making the safety of their use a significant issue [8,9]. In addition, most of these chemotherapies are included in the list of “hazardous to handle” drugs established by the National Institute for Occupational Safety and Health (NIOSH) in the United States in 2004 and whose latest version dates from 2016 [10]. Occupational chemotherapeutic exposure induces three main risks: immediate organ toxicity, impairment of reproductive functions [11], and cancer pathology [12]. Therefore, despite all modern medicine’s efforts, the mortality rate is still increasing worldwide.

The natural world abounds with a multitude of species and plant diversity. The diversity of chemical structures and their relatively low toxicity make natural products of plant origin a promising source for developing new anti-cancer therapies that are more effective and capable of targeting multiple characteristics of cancer. In Benin, several testimonies have been reported regarding treating diseases of microbial and viral origin with the organs of cassava. This plant is highly produced in Benin and used mainly in human food in various artisanal and industrial forms [13]. The root is consumed as a food product (in the form of “*gari*”, “*tapioca*”, “*lafun*”, and “*agbeli*”) and is a good source of starch and biofuel. Besides their nutritional importance, cassava leaves and roots are also used to treat several diseases, such as diabetes, rheumatoid arthritis, cell aging, and cardiovascular diseases, including atherosclerosis [14]. These organs contain bioactive molecules like vitamins C and A and secondary metabolites such as flavonoids, saponins, steroids, and cyanogenic glycosides [14]. Besides these bioactive molecules, cassava contains amygdalin, which has proven therapeutic effects.

Amygdalin is a popular cyanogenic disaccharide [15]. This compound is attributed to the high therapeutic effects provided by several authors, such as the anti-inflammatory and analgesic actions of neurodegenerative diseases [16]. It also treats asthma, bronchitis, emphysema, leprosy, and diabetes [17]. Extracted from apricot (*Prunus armeniaca*), amygdalin inhibits breast, lung, and bladder cancer cells [18]. Regarding the toxicity of this molecule, low and medium doses (50 and 100 mg/kg) of amygdalin administered orally do not induce toxicity in mice [19]. While amygdalin in a high amount (200 mg/kg) can induce toxicity, causing adverse effects on the oxidative balance of liver tissues and a pronounced impact on mouse histopathology [19]. So, the highest dose of amygdalin that does not cause unacceptable side effects in mice, rabbits, and dogs is 3 g/kg for intravenous and intramuscular injections and 0.075 g/kg for oral administration. Moreover, the maximum tolerated dose of amygdalin injected intravenously in humans is approximately 0.07 g/kg [20]. Because of these multiple bioactive molecules, studies on quantifying and analyzing the therapeutic effects of amygdalin extracted from cassava remain to be explored.

In Benin, cancer treatment is not affordable for all population groups. Moreover, cassava is a plant with a prominent place in the population’s diet and contains bioactive molecules. Furthermore, cassava is a plant in several varieties whose organs are used. These organs occupy a place of choice in the diet of the Beninese population and are full of bioactive molecules. Do we wonder which plant varieties, organs, or derivatives contain more amygdalin? The objectives of the present study were (i) to identify the major groups of secondary metabolites of cassava varieties and (ii) to evaluate the efficacy of amygdalin extracted from three of the most consumed cassava varieties in Benin. Specifically, the aim was to assess the antioxidant, antimicrobial, anti-inflammatory, and anticancer activities of amygdalin extracted from three cassava varieties in Benin.

## 2. Results

### 2.1. Phytochemical Screening

Qualitative tests revealed the presence of various secondary metabolites such as flavonoids, saponosides, steroids, tannins, leuco-anthocyanins, glycosides, alkaloids, coumarins, and cyanogenic derivatives (Table 1). The results showed that the three cassava varieties do not contain triterpenes.

### 2.2. Extraction Yields

The results of the extraction yield of the various cassava organs are presented in Table 2. The best extraction yield among the organs of the cassava samples was obtained with the second skin, followed by the pulp, regardless of the variety. The second skin of the BEN variety dried in the sun gave the best yield (18.21%). The lowest yield was obtained with the first skin (0.95%) of the RB variety dried in the shade and sun. Furthermore, extracting the leaves of the three varieties gave the exact yield at extraction, which is 1.5%.

### 2.3. Phytochemical Constituents of Three Cassava Varieties

The contents of polyphenolic compounds, flavonoids, and total tannins of the samples of the three varieties (BEN, MJ, and RB) of cassava are presented in Table 3. The MJ variety has a higher flavonoid content (129.36 ± 9.22 µgEQ/100 mg) compared to the BEN (110.96 ± 1.18 µgEQ/100 mg) and RB samples (125.20 ± 2.77 µgEQ/100 mg). The total content of polyphenols varied significantly (*p* < 0.05). The RB extract had the highest total tannin content (0.54 ± 0.03 mgAAG/g), while the BEN (0.35 ± 0.07 mgAAG/g) extract had the lowest.

### 2.4. Variation of the Amygdalin Content in Cassava Varieties Organs

Amygdalin was identified by the HPLC method in methanolic extracts by comparing the absorption chromatogram of the standard, which is the standard of amygdalin (Sigma Aldrich (St. Louis, MO, USA)—TCH—10050-5G; purity ≥ 97%) at 30 µg mL^−1^ and that of the samples. This chromatogram showed (Figure 1) its peak at a retention time equal to 4.1 min.

#### 2.4.1. Amygdalin Content in Cassava Stem Organs

The amygdalin content of cassava stems according to cassava varieties is presented in Figure 2. Six (06) samples were submitted for amygdalin determination. Amygdalin is present in all samples submitted for analysis. Note that the young, fresh stems of yellow cassava are “MJT_F”.

#### 2.4.2. Amygdalin Content in Cassava Leaves

The amygdalin content of cassava leaves according to variety is presented in Figure 3. Six (06) samples were submitted for the determination of the molecule. Amygdalin is also noted in all the pieces submitted for analysis. Here, the contents found are a function of the variety and the drying because the contents of amygdalin in the fresh leaves are superior to those obtained in the leaves dried in the shade. Manioc Fresh Leaf Yellow “MJF_F”, followed by BEN Fresh Leaf “BF_F”, contained the highest levels of amygdalin (9251.74 µg and 7611.98 µg, respectively). On the other hand, the variety RB Leaf dried in the shade “RBF_OM” contains the lowest amygdalin content (430.96 µg).

#### 2.4.3. Amygdalin Content in Cassava Derivatives

The amygdalin content of cassava derivatives is presented in Figure 4. Eight samples were subjected to amygdalin determination. Among these derivatives, *Agbéli* has the highest amygdalin content (401.56 µg/10 g of sample). It should be remembered that *Agbeli* is fermented cassava starch. The boiled flesh also had interesting amygdalin contents (from 30 to 138.38 µg/10 g of sample). All the contents found are very variable from one derivative to another. This variation can be explained by the technology used to obtain each product. Furthermore, the *Gari* derivative has the lowest amygdalin content (28.09 µg/10 g of sample), while the *Tapioca* derivative has no readable value. It should be noted that, of all the products studied, these two derivatives are manufactured by heating at a very high temperature. The amygdalin content in *Gari* is low compared to that found in fresh roots.

### 2.5. Antioxidant Activity

The variations of the antioxidant power by the DPPH method of amygdalin extracted from different organs (flesh, 1st and 2nd skin) of three cassava varieties are presented in Table 4. In general, it is observed that the antiradical power varies from one extract to another and that the highest values are obtained with BEN 1st skin (PI: 95.38 ± 0.07%; IC_50_ < 0.19 µg mL^−1^) followed by MJ flesh (PI: 95.11 ± 0.18%; IC_50_ =0.5 ± 0.22 µg mL^−1^). The flesh of BEN and 1st skin MJ (respectively with PI: 94.71 ± 0.07%; IC_50_ < 0.19 and PI: 93.77 ± 0.54%; IC_50_ = 0.25 ± 0.07 µg mL^−1^) for sun-dried samples. There was no significant difference (*p* > 0.05) between these values. While the lowest values are obtained with MJ 2nd skin and RB flesh (PI: 85.12 ± 0.67%; IC_50_ =4.6 ± 1.97 µg mL^−1^ and PI: 87.28 ± 1.25%; IC_50_ =3 µg mL^−1^). The percentages of inhibition are well above 50%.

For the shade-dried samples (Table 5), the highest average free radical scavenging power through DPPH is obtained with MJ 2nd skin (PI: 92.10 ± 0.16%) with an IC_50_ equal to 0.5 µg mL^−1^ followed by BEN 2nd skin (PI: 91.37 ± 0.18%) with an IC_50_ = 2.75 µg mL^−1^. The lowest value of antiradical power is obtained with BEN flesh (PI: 85.88 ± 0.28; IC_50_ = 9.6 ± 0.56 µg mL^−1^) and RB flesh (PI: 88.93 ± 0.12%; IC_50_ = 17.25 ± 0.35 µg mL^−1^). The reference molecule used as a standard (quercetin) has a lower free radical scavenging power (PI: 82.35 ± 1.86%) than all samples and an IC_50_ of 5.75 µg mL^−1^.

The variation in antioxidant power by the DPPH method of the three cassava varieties leaf extracts is presented in Table 6. It can be seen that the IC_50_ of the BEN and MJ varieties are almost equal (3.79 and 3.02 µg mL^−1^, respectively) and are lower than those of the RB variety (8.11 µg mL^−1^). Note that the IC_50_ of the reference molecule (ascorbic acid, IC_50_ = 1.11 µg mL^−1^) is lower than the IC50 of amygdalin in all three varieties. There is no significant difference (*p* > 0.05) between the IC_50_ values obtained (µg mL^−1^) with the reference molecule and the amygdalin of the leaves of the BEN and MJ varieties. The RB extract shows the highest reducing power.

The efficiency of the extracts in reducing iron was determined. Table 6 shows that the highest declining powers were recorded with the extracts from the leaves of varieties BEN and MJ (IC_50_ of 0.63 ± 0.04 µg mL^−1^ and 0.69 ± 0.03 µg mL^−1^, respectively) compared to the reducing power of the leaves of variety RB (IC_50_ of 0.52 ± 0.04 µg mL^−1^). The comparative analysis between these values shows a significant difference between MJ-BEN and RB (*p* < 0.05). The extracts of BEN and MJ varieties are more active in the FRAP test than those of the RB variety. Thus, the presence of the reductants in the extracts caused the reduction of the Fe^3+^ ion (complexed) to the Fe^2+^ ion.

### 2.6. Larval Cytotoxicity of Amygdalin Extracted from Cassava Varieties

The cytotoxicity of the extracts was evaluated on *Artemia salina* larvae. The equation of the regression line was used to determine the concentrations of ethanolic extracts of the leaves that caused the death of 50% of the larvae (LC_50_) previously introduced. The ethanolic extracts of the leaves of the three cassava varieties (BEN, MJ, and RB) showed no toxicity in the tested larvae. Nevertheless, the CL_50_ of the RB and MJ extracts were almost the same (12.73 and 12.32 mg mL^−1^). These values are higher than the BEN variety’s (Table 7). Of all the extracts tested, RB extract recorded the highest larval losses at high concentrations (10/16 dead larvae at 100 µg mL^−1^) (Figure 5).

### 2.7. Anti-Inflammatory Activity of Cassava Leaf Extracts

The 5% formalin-induced acute mouse hind paw edema model was used. The edema size was measured regularly every hour for 5 h, and the percentages of increase and inhibition were calculated. The results are shown in Figure 6 and Figure 7.

Administration of ethanolic extracts at 100 mg/kg and acetylsalicylic acid at 50 mg/kg prevents edema in treated mice compared with control mice that received physiological water alone. The prevention is highly significant (*p* < 0.0001) with BEN extract from the third to the fifth hour. An increase in the volume of the mice’s paws was observed in the control group, with a maximum of 55.16% at the first hour compared to a maximum of 48.88% in the lot that received aspirin and a maximum of 47.94% in the lot that received the BEN extract, always at the same hour. On the other hand, the batch of mice that received MJ extract showed a maximum of 43.61% in the second hour. Suppose the percentage increase in edema is more or less stable (between 53.69% and 55.16%) in control mice compared to mice treated with BEN extract. In that case, these percentages decrease considerably from the third to the fifth hour. In mice treated with MJ extract, the rate of edema increase decreases progressively from the second to the fifth hour. The batch treated with aspirin showed a first decrease between the first and second hour (48.88% to 41.16%) and a second decrease between the fourth and fifth hour (44.19% to 40.09%) (Figure 6).

Moreover, the percentages of edema inhibition increase with time. There is no significant difference (*p* > 0.05) between the anti-inflammatory effect of the extracts at 100 mg/kg and that of aspirin at 50 mg/kg from the first hour to the fifth hour. Better yet, like the reference molecule (acetyl salicylic acid), our extracts have a more crucial anti-inflammatory activity in the second phase of the inflammatory process.

This while better inhibition of inflammatory paw edema was observed at the fifth hour in all treated mice, with inhibition percentages ranging from 27.89% (BEN extract), 25.20% (aspirin), and 21.77% (MJ extract) (Figure 7).

### 2.8. Anticancer Activity of Amygdalin Extracts from Cassava Variety BEN

#### 2.8.1. Biochemical Analysis

The variation in biochemical parameters is presented in Table 8. This table shows that the mean value of the creatinemia levels of the rats in group 2 is very high (19.57 mg L^−1^) compared to the other batches. This high value indicates the onset of lesions and kidney dysfunction (renal failure) in the rats of batch 2. This finding is due to the treatment received by the rats in this batch, which consisted exclusively of DMH injections during the entire period. DMH, therefore, acted adversely on the liver and kidneys of the rats in this batch.

Overall, the values of the biochemical parameters show a clear difference between the different batches of rats. Primarily, there is a big difference between the parameter values of the rats in group 2 and the parameter values observed in all other remaining batches (R1, R3, R4, and R5). This diversity of values is observed in all parameters except those of monocytes. The variation of all the values of the biochemical parameters is highly significant (*p* < 0.001) when comparing one batch to another, except for monocytes (*p* > 0.005), as observed below (Table 8). In addition, after reaching the mean values of biochemical parameters made two by two between the batches of rats, it is found that the mean values of the rats in group 2 are broadly higher than the mean values of the other batches. On the other hand, there is a clear difference between the values of the biochemical parameters of the rats in the control lot (R0), which did not receive any treatment, compared to the values of the other groups (R1 to R4). Still, these values of the lot show similarities with the values of lot 5, which received treatment with the reference molecule (5-fluorouracil). The values of the latter’s biochemical parameters are similar to those of the rats (R1, R3, and R4) that received the treatment with amygdalin extracts (preventive and curative). This finding reflects the similarity in the modes of operation of 5-fluorouracil and amygdalin.

#### 2.8.2. Histological Analysis

##### Observation of the Colon

The effects of DMH and treatments on the colon of rats are shown in Figure 8. In the DMH-treated group (b), the colonic mucosa shows altered Lieberkühn’s glands (G) with areas of cellular infiltrates (CI), suggesting unchecked cellular proliferation. In groups 1, 4, and 5, the mucosa has its typical architecture as in normal rats (control group), with Lieberkühn’s glands (G) presenting their aspect of simple tubular glands with an abundance of caliciform cells (arrows). In group 3, the glandular appearance is more or less standard, with slightly fewer caliciform cells (arrows). The caliciform cells seem to disappear from the glands. DMH is at the root of the unchecked cell proliferation observed in the rat colon.

##### Observation of the Liver

Apart from the observations on the colon, the possible effects of DMH on the liver were examined. The histology of the liver of experimental Wistar rats is presented in Figure 9. This figure shows that the liver parenchyma did not show any visible atypia in the different batches. The hepatocytes (arrows) are well organized in cords around the centrilobular veins (VC). Venous sinusoids (S) are visible between the hepatocyte cords. This explains the mechanism of action of DMH, a molecule that acts primarily on the colon where its binding sites are present, allowing it to create damage in the colon. This damage to feedback can have harmful consequences for the liver later on. These observations show that DMH has not yet had any detrimental effect on the livers of treated rats.

##### Observation of the Kidney

The effects of DMH and treatments on the kidneys of rats are shown in Figure 10. In the untreated positive control group (Batch 2), the renal parenchyma showed cellular debris (arrows) in some tubular lumens, indicating cellular injury. In the other groups, the appearance of the parenchyma is typical, with distinct glomeruli (G) and renal tubules (RT). The tubular lumens are visible. These observations show that only DMH acts negatively and moderately on the kidneys of treated rats. In the other groups, the appearance of the parenchyma is typical, with distinct glomeruli (G) and renal tubules (RT). The tubular lumens are visible. These observations show that only DMH acts negatively and moderately on the kidneys of treated rats.

## 3. Discussion

The phytochemical study characterized the presence of glycosides, flavonoids, saponosides, steroids, tannins, leuco-anthocyanins, coumarins, and cyanogenic derivatives in ethanolic extracts of leaves of cassava varieties (BEN, RB, and MJ). The content of these secondary metabolites varies from one type to another. These same metabolites were found in the leaves of a cassava variety harvested in Côte d’Ivoire [21]. Cyanogenetic compounds were detected in the sample powders. These are anti-nutritional factors that contribute to cassava’s low protein and mineral content. However, this toxicity can be reduced, or even eliminated, in the finished ready-to-eat products obtained after various transformations.

Determining total phenolic compounds, flavonoids, and total tannins showed a variation in their contents from one variety to another. The ethanolic extracts from MJ and BEN varieties are much more concentrated in total flavonoids. In contrast, these varieties show a low level of total phenols and total tannins compared to the extract from the RB variety. Different studies have shown that external factors (geographical and climatic factors), genetic factors, but also the degree of maturation of the plant, and storage time strongly influence the content of secondary metabolites [22].

Amygdalin was determined in 39 samples of cassava produced in southern Benin. Pure methanol (99.8% for HPLC. LABO CHEMIE PVT. Ltd., Mumbai, India) was used for the HPLC assay, which allowed a good quantification of amygdalin in each sample. Because methanol is an excellent mobile phase for amygdalin separation by HPLC in less than 50 mn [23]. The recovery rate found in this study was 98% (with a correlation coefficient of R = 0.99). This result corroborates a study [24] that separated amygdalin within 15 mn in almond seeds and food products in the UK. The result is similar to that of [25], who worked on artemisinin grown in Benin. These authors also found a recovery rate of 98%.

The content of amygdalin varied significantly (*p* < 0.05) between different organs and derivatives of cassava. Among the samples analyzed, amygdalin was more abundant in young stems and fresh leaves of cassava, with a content of 11,142.99 µg 10 g^−1^ and 9251.14 µg 10 g^−1^, respectively. While, among cassava derivatives, the *Agbeli* derivative was more concentrated in amygdalin with a range of 401.56 µg/10 g. These contents are lower than the 141.000 µg 10 g^−1^ and 155.000 µg 10 g^−1^ obtained, respectively, with almond seeds and Mongolian almonds reported in the studies of [26] in China. High levels were also received with green plum, apricot, black plum, peach, red cherry, and black cherry, as reported by [24] (17.5 mg g^−1^, 14.4 mg g^−1^, 10 mg g^−1^, 6.8 mg g^−1^, 3.9 mg g^−1^ and 2.7 mg g^−1^, respectively). The differences observed with these results can be explained by variations in the plant species studied, climatic and environmental factors, and the soil types exploited to grow these plants. Geographic environment and genomic differences could greatly influence amygdalin biosynthesis and accumulation [26].

The antioxidant activity of plant extracts containing phenolic compounds is due to their ability to act as hydrogen or electron donors and scavenge free radicals. The DPPH test is commonly used to prove the antioxidant capacity of fractions and isolated pure compounds to act as hydrogen atom donors [27]. The results obtained in our case study show that the sun-dried cassava organs proved to be the more potent scavengers of DPPH radicals than the shade-dried organs with IC_50_ values, including one below 0.19 mg/mL and others of 0.25, 0.5, 0.75, and 2.35 mg mL^−1^. These values are also more attractive than the leaves of the three varieties. They are also better than the IC_50_ of ethanolic extracts of cassava stems (0.518 and 0.616 mg mL^−1^) reported by [28] in China and then those of methanolic extracts of peels, which contain a yellow-fleshed variety of cassava (425 and 234 μM TE g ^−1^) reported in the studies conducted by [29] in Nigeria. At the same time, authors [30] said that water yam and dasheen (*Colocasia esculenta*) had the same high percentage of DPPH inhibition activity, with 95.83% and 93.41%, respectively. The high values of antioxidant activity can be attributed to high levels of phenols and flavonoids coupled with other compounds such as phenylpropanoids and anthocyanins [31]. Furthermore, leaf ethanolic extracts of leaves showed significantly higher DPPH radical scavenging activity than cassava leaf stem extracted with acidified methanol, simple methanol, and acetone, as reported by [32] in India. These observed differences may be related to the different phytochemical compositions of the plant parts and the extraction solvents. Indeed, according to [33], the antioxidant capacities of the extracts have a strong relationship with the solvent used, mainly due to the different antioxidant potentials of compounds of different polarities.

Furthermore, the reducing power value of quercetin, chosen as the reference molecule, is lower (82.35%) than the reducing power values of most of our samples. Therefore, the organs of all sun-dried varieties are more active in the DPPH test than those of shade-dried varieties. This could be explained by the variation in the level of secondary metabolites contained in the different organs of each cassava variety and the temperature related to the drying methods of these organs.

Furthermore, extracts from BEN and MJ leaves are more active in the FRAP test than extracts from other organs. The reducing power of these extracts is undoubtedly due to the presence of hydroxyl groups in the phenolic compounds, which serve as electron donors. Thus, antioxidants are considered to be reducers and inactivators of oxidants [34].

This study also assessed possible risks to the population using the leaves of the three cassava varieties. The larval cytotoxicity curves showed that larval mortality increases with concentration, and referring to the toxicity scale established by [35], all the LC_50_ values of our ethanolic extracts are higher than 0.1 mg/mL, a value above which the section is considered not to present toxicity. These results show that the ethanolic extracts of the leaves of the three cassava varieties (RB, BEN, and MJ) are biologically active at a dose of 100 mg/mL and are non-cytotoxic. Therefore, these leaves’ medicinal and food uses do not present any short- or long-term intoxication risk to the populations. It should be recalled that this study showed the presence of secondary metabolites, preeminent chemical groups such as flavonoids and phenols. The detected components have various therapeutic properties, such as the astringent effects of tannins and the anti-inflammatory and anti-allergic effects of flavonoids. Besides their antioxidant power, they are anti-ulcerous, antispasmodic, antisecretory, and antidiarrheal [36]. They are also endowed with aphrodisiac virtues [37].

The results obtained from the anti-inflammatory tests show that ethanolic extracts of the leaves of the three cassava varieties at 100 mg/kg appreciably reduce the edema induced by formalin. Injection of 5% formalin into the paws of mice provoked an almost immediate inflammatory response manifested by the appearance of classical signs of acute local inflammation, such as redness, pain, heat, and edema, in all four experimental groups. This inflammation begins with a phase that lasts about 1 h 30 min after injecting 5% formalin and is triggered by the production of serotonin, histamine, and bradykinin. Formalin causes local inflammation when injected into the fascia of the sole [38], as does carrageenan [39]. The second phase, which occurs after the second hour until the fifth hour, is due to the biosynthesis of prostaglandin [40] associated with leukocyte migration to the inflamed area [41]. The cause of this inflammatory response is a tissue injury that induces the synthesis of histamine, prostaglandins, leukotrienes [42], PAF (p1aqueta activating factor), cytokines, NO (nitric oxide), and TNF (tumor necrosis factor) [43]. According to [44], these mediators promote vasodilation, which causes redness and heat at the site of inflammation.

In addition, the 100 mg/kg ethanolic extract of BEN leaves reduced edema more significantly (*p* < 0.0001). On the other hand, there was no significant difference (*p* > 0.05) between the anti-inflammatory effect of the two ethanolic extracts (BEN and MJ) and that of the 100 mg/kg standard. It can be deduced that BEN and MJ extracts act in the same way as salicylic acid. Studies have shown that salicylic acid, used as a standard, works in the second phase of inflammation while inhibiting the synthesis of these different mediators [45]. By inhibiting the production of prostaglandins through the inhibition of cyclooxygenase (COX2), it will limit the lowering of the pain threshold, hence its analgesic action, as well as inflammatory reactions, hence its antipyretic activity [46,47]. This presages the same pharmacological responses with our extracts. These results suggest that amygdalin extracted from cassava leaves has an effect that opposes the action of endogenous pro-inflammatory mediators. This action would be exerted more on cyclooxygenase, the enzyme responsible for synthesizing prostaglandins [48].

The anticancer activity test shows that subcutaneous injection of DMH into treated rats induces high levels of biochemical parameters and, consequently, the observation of tumors in the colons of the rats that received it. 1,2-dimethylhydrazine is an effective carcinogen for the induction of colon and rectal tumors in rats and mice by systemic subcutaneous or intraperitoneal injections [49]. In addition, researchers studied the efficacy of DMH in female mice. They found that 83% of the mice developed visible tumors, and many had them primarily in the distal part of the colon [50]. This justifies the result observed in the colon of batch 2 rats (R2 treated with DMH). Recall that the procarcinogen DMH, after a series of metabolic reactions, finally reaches the colon, where it produces the ultimate carcinogen and an imbalance in the production of reactive oxygen species (ROS), which alkylate DNA and initiate the advent and development of colon carcinogenesis [51]. The preneoplastic lesions and histopathological observations of DMH-induced colon tumors can provide a typical understanding of the disease in rodents and humans. Therefore, the interpretation of histopathological observations revealed cellular abnormalities mainly in the colon of these rats from batch 2. On the other hand, the absence of visible tumors in the colon, liver, and kidney of rats of batches 1, 3, 4, and 5 (batches that received amygdalin from BEN leaves and 5-fluorouracil) attests that the ethanolic extracts (amygdalin), as well as the reference molecule, affected DMH-induced cell proliferation.

Previous studies have reported that high levels of amygdalin ingested directly can be toxic to humans. Amygdalin comprises two glucose molecules, benzaldehyde and hydrogen cyanide, and can exist as two epimers, R and S [52]. The R-amygdalin is the natural amygdalin, and the S-amygdalin is called neo-amygdalin. Beta-glucosidase stored in plant cell compartments is also present in the human small intestine [53] and degrades amygdalin to prunasin, mandelonitrile, glucose, benzaldehyde, and hydrogen cyanide. Hydrogen cyanide (HCN), benzaldehyde, prunasin, and mandelonitrile can be absorbed into the lymphatic and portal circulations [54]. The anticancer activity of amygdalin is thought to be related to the cytotoxic effects of enzymatically released HCN and unhydrolyzed cyanogenic glycosides [55]. But low and medium doses (50 and 100 mg kg^−1^) of amygdalin administered orally do not induce any toxicity, while high doses of amygdalin (200 mg kg^−1^) can cause toxicity [19].

## 4. Materials and Methods

### 4.1. Chemicals

The extraction solvents, phosphate-buffered saline (PBS), methanol (CH_3_OH), and ethanol (C_2_H_5_OH), were obtained from Sigma-Aldrich Chemical Company (St. Louis, MO, USA). The 2,2-diphenyl-2-picrylhydrazyl (DPPH), potassium hexacyanoferrate [K_3_Fe(CN)6], trichloroacetic acid (C_2_HCl_3_O_2_), gallic acid (C_7_H_6_O_5_), ascorbic acid (C_6_H_8_O_6_), quercetin (C_15_H_10_O_7_), iron chloride (FeCl_3_), Folin–Ciocalteu reagent (FCR), anhydrous sodium carbonate (Na_2_CO_3_), aluminum chloride (AlCl_3_), and potassium acetate (CH_3_CO_2_K) were of analytical grade. 1,2-dimethylhydrazine (DMH) was purchased from Macklin (Shanghai Macklin Biochemical Co., Ltd., Shanghai, China). Standard amygdalin was purchased from Sigma-Aldrich (Sigma-Aldrich, Co., 3050 Spruce Street, St. Louis, MO, USA), and 5-fluorouracil was provided by Sigma-Aldrich, Germany (St. Louis, MO, USA). Biochemical analysis kits were BIOLABO diagnostic kits.

### 4.2. Plant Material and Sample Collection

The plant material consists of the peelings (1st and 2nd skin) of the root, pulp (core), stem, and leaves of three different varieties. These are the BEN, RB, and yellow manioc (MJ) (Table 9). The samples were collected in the commune of Toffo, located in the Atlantic Department. These samples were rinsed lightly with water to remove dust.

### 4.3. Methods of Drying Cassava Organs

#### 4.3.1. Shade Drying in the Laboratory

Organs such as the leaves, pulp, and stems of cassava were dried in the shade. This drying consisted of carefully cleaning the bench. On this bench, the different samples were dried under air conditioning at 16 °C for six (6) days.

#### 4.3.2. Traditional Sun Drying

The organs, such as the flesh and the cassava root’s second skin, were dried in the sun. This drying involves spreading an empty and clean PET bag on the ground. In this bag, the different samples are spread out in the sun for six (06) days.

The dried samples were ground to powder using a mill (IKA Bro-03-PATACE-018, Königswinter, Germany).

### 4.4. Animal Material and Acclimatization Conditions

This study was carried out within the framework of a doctoral thesis. The animals used were albino mice and Wistar rats of EOPS (exempt from a specific pathogenic organism) health status, age approximately eight, and weighing between 150 g and 200 g. The animals were housed in polypropylene cages integrated with water pots under hygienic conditions with standard rat food and free access to water. After two weeks of acclimatization at a constant temperature of 22 ± 2 °C under a 12/12 light/dark cycle, the rats were divided into batches for the different tests. The body weight of the mice and rats was recorded at the experiment’s beginning and end. The protocol received the favorable opinion of the scientific committee of the Doctoral School of Life and Earth Sciences (ED-SVT) of the University of Abomey-Calavi (UAC).

### 4.5. Phytochemical Screening of Cassava Organs

The phytochemical screening carried out on the organ powder is based on differential precipitation and coloring reactions using the method of [56]. This study identified the leading chemical groups (tannins, flavonoids, coumarins, terpenes, alkaloids, etc.) contained in the plant material.

#### 4.5.1. Preparation of Extracts

Dried powdered cassava organs were subjected to reflux conditions using the solvent ethanol at 78.5 °C according to the method of [57]. 20 g of each ground organ was mixed with 200 mL of ethanol. Each mixture was boiled under reflux for 40 min. After refluxing, the mixture was subjected to ultrasonication for 15 min at 30 °C and then completely filtered using Whatman No. 1 filter paper, and the solvents were evaporated using a rotary evaporator. After drying the filtrate in the oven, 10 mL of diethyl ether was added to each filtrate and stirred for 1 min at room temperature (22 °C) to obtain the precipitate. Diethyl ether was evaporated overnight in a fume hood. The crude extract was collected in Petri dishes, placed in the oven for drying, and then scraped into sterile glass pillboxes for excellent storage at 4 °C for biological testing. The extraction yield is the ratio of dry extract mass obtained to the group of plant material processed [58].

#### 4.5.2. Determination of Total Phenols

Total phenolics are quantified in leaf extracts of each cassava variety (BEN, RB, and MJ) using the Folin–Ciocalteu method described by [59] and adapted to an in-house method developed by [60]. A quantity of 10 µL of concentrated extract at 100 mg/mL (*w*/*v*) was introduced into the wells of 96-well plates with three replicates. Twenty-five microliters of Folin–Ciocalteu reagent (50% *v*/*v*) was added and incubated for 5 min at room temperature. Then, 25 µL of sodium carbonate (Na_2_CO_3_) at 20% (*w*/*v*) was added and then reduced to 200 µL with distilled water per well. Blanks were prepared by replacing the reagent with distilled water to correct for interfering compounds. Incubate for 30 min, and absorbance is read at 760 nm on a multi-well plate reader. Gallic acid (0–500 µg/mL) is used as a standard, and results are expressed as micrograms of gallic acid equivalent per 100 g of extract.

#### 4.5.3. Determination of Total Flavonoids

Total flavonoids in each sample were quantified by the aluminum trichloride method described by [61] and adapted to 96-well plates by [60]. It consists of adding and mixing 100 µL of methanolic AlCl_3_ (2%) to 100 µL of the appropriate dilution of the extract solution. Incubate for 15 min, then read the absorbance at 415 nm with the “Gen5” software using an Epoch Biotech multi-plate spectrophotometer connected to a computer against a blank (mixture of 100 µL of methanolic extract solution and 100 µL of methanol) against a standard. Values were compared to a calibration curve for quercetin (0–500 µg/mL) at R^2^ = 0.99. Flavonoid content is expressed as micrograms of quercetin equivalents per 100 g of extract.

#### 4.5.4. Determination of Hydrolyzable Tannins

The hydrolyzable tannins were determined according to the method of [62]. Indeed, 1 mL of each extract (5 mg/mL) was mixed with 3.5 mL of reagent (ferric chloride FeCl_3_ 10^−2^ M in hydrochloric acid HCl 10^−3^ M). The absorbance of the mixture was measured (UV spectrophotometer, Shimadzu) at 660 nm after 15 s of incubation. The content of hydrolyzable tannins T (%) was determined according to the formula below:T (%) = (A × PM × V × DF)/ε mole × P
where A = absorbance, PM = weight of gallic acid (170.12 g/mol), V = volume of extract, DF = dilution factor, ε mole = 2169 (gallic acid equivalence constant), and P = weight of the extract.

### 4.6. Method for the Determination of Amygdalin by HPLC

#### 4.6.1. Extraction and Analysis of Amygdalin from Cassava Samples/Derivatives

Amygdalin from cassava organs and products was extracted according to the method described by [63] with some modifications. A quantity of 10 g of each sample was macerated in 150 mL of absolute methanol (Sigma Aldrich) for 15 min at 30 °C. Vacuum filtration was performed using a buchniii system and then concentrated in the rotavapor at 40 °C. The resulting filtrate was washed with 25 mL of diethyl ether and allowed to decant for 3 h. After decantation, the crude extract was obtained by removing the supernatant.

Amygdalin was separated and quantified with the HPLC-VWR-HITACHI system using the following analytical conditions: C18 column (LiChrospher 100 (5 μm), 100 × 4.6 mm Merck Chemicals, Darmstadt, Germany); mobile phase: methanol/water (20/80, *v*/*v*); flow rate: 0.7 mL/min; analysis time: 7 min; detection wavelength: 210 nm.

#### 4.6.2. Quantification Procedure

To validate the method developed by the precision criteria, we performed intra-day and inter-day replicates. Precision was determined for the different concentrations of amygdalin three times on the same day for intra-day accuracy and three times on four other days for inter-day precision. Three injection tests were performed for each dilution, and the means of the areas obtained under the curves were calculated. These means and their standard deviations were used to calculate the coefficients of variation (CV). The extraction yield was determined from 10 g of cassava organs or derivatives with the addition of 1 mg of pure amygdalin. The formula gives the recovery rate (RR):$$\mathrm{RR}\left(\%\right)=\frac{\left(A-B\right)}{C}\times 100$$
where A: amount of amygdalin extracted from cassava samples mixed with pure amygdalin; B: amount of amygdalin in cassava samples; and C: amount of amygdalin added.

After calibration, the method was applied to the samples. The expression of the content (T) of amygdalin in µg in 100 g of sample is T = 10 q with the amount of amygdalin in 10 µL of injected extract.

#### 4.6.3. Validation of the Amygdalin Identification Method

Amygdalin was identified by the HPLC method in methanolic extracts by comparing the absorption chromatogram of the standard, which is the standard of amygdalin (Sigma-Aldrich—TCH—10050-5G; purity ≥ 97%) at 30 µg mL^−1^ and that of the samples. This chromatogram showed its peak at a retention time equal to 4.1 min.

### 4.7. In Vitro Evaluation of the Antioxidant Activity of Different Extracts

#### 4.7.1. Reduction of the DPPH Radical

It was evaluated according to the method described by [64]. Thus, starting from 200 μg/mL of the extract solution, a series of 10 successive dilutions (to 1/2) of each extract was made with methanol to have 100 μL of volume. Then 100 μL of DPPH (100 μg/mL in methanol) was added to all dilutions in a cascade. At the same time, a negative control was prepared by mixing 100 μL of methanol with 100 µL of the methanolic DPPH solution. The optical density reading was taken against a blank prepared for each concentration at 517 nm after 20 min of incubation in the dark at room temperature. A solution of a standard antioxidant represents the positive control, ascorbic acid, whose optical density was also measured under the same conditions as the samples. For each concentration, the test was repeated 3 times. The anti-free radical activity is estimated according to the following equation:$$\mathrm{PI}\left(\mathrm{DPPH}\right)=\frac{\mathrm{DO}\left(\mathrm{control}\right)-\mathrm{DO}\left(\mathrm{extract}\right)}{\mathrm{DO}\left(\mathrm{control}\right)}\times 100$$
where PI (DPPH) = percentage of inhibition of DPPH; DO = optical density

#### 4.7.2. Evaluation of the Reducing Power of Extracts

The ability of the extracts to reduce Fe^3+^ was assessed using the ferric reducing antioxidant power (FRAP) method described by Dieng et al. [65] with a slight modification. Briefly, 1 mL of the extracts was mixed with 2.5 mL of phosphate buffer (0.2 M, pH 6.6) and 2.5 mL of 1% potassium ferricyanide [K_3_]. Potassium ferricyanide [K_3_Fe(CN)_6_]. After 20 min of incubation at 50 °C, 2.5 mL of trichloroacetic acid (10%) was added to the mixture, mixed, and centrifuged for 10 min (3000 r/t). Then, 2.5 mL of the supernatant was mixed with 2.5 mL of distilled water and 0.5 mL of ferric chloride (0.1%). The absorbance was read at 700 nm. Ascorbic acid was used for the calibration curve and rutin (200 µg/mL) as the standard. Ferric iron (Fe^3+^) reducing activity was determined as follows: ascorbic acid equivalents (mmol ascorbic acid/g extract). The antioxidant activity related to the reducing power of the extracts is expressed as reducing power (RP) using the following formula:$$\mathrm{PR}=\frac{\mathrm{DO}\left(\mathrm{extract}\right)-\mathrm{DO}\left(\mathrm{blanc}\right)}{\mathrm{DO}\left(\mathrm{extrait}\right)}\times 100$$

DO: optical density.

The probit method determined the IC_50_ (concentration that inhibits 50% of DPPH or reduces 50% of Fe^3+^).

### 4.8. Evaluation of the Cytotoxicity of Extracts

The larval toxicity of the extracts was evaluated according to the method described by [66]. Larvae were obtained by hatching 10 mg of freeze-dried eggs of Artemia salina (ARTEMIO JBL GmbH D-67141, Neuhofem, Germany) under continuous agitation in one liter of seawater for 72 h. Thus, starting from a stock concentration of 100 mg/mL extracts, a range of 10 concentrations was made with sterile distilled water following half dilutions of ½ reason in test tubes numbered from T1 to T10. Then, 1 mL of seawater containing 16 live larvae was added to all lines. The number of surviving larvae was counted under the microscope after 24 h of incubation. The lethal concentration of 50 (LC50) for each extract was determined from the regression line obtained from the curve representing the number of surviving larvae as a function of the extract concentration. The results were interpreted using the correlation grid, associating the degree of toxicity with the LC_50_ proposed by [34].

### 4.9. In Vivo Evaluation of the Anti-Inflammatory Activity of Leaf Extracts from Three Varieties of Cassava

To highlight the anti-inflammatory activity of the studied cassava extracts, an experimental model of acute inflammation of the mouse paw induced by 5% formalin was used following the protocol used by [67]. The experiments were performed in male and female Swiss albino mice weighing 25 g and 43 g. They were fed pellets containing 29% protein and received water ad libitum. They were maintained at a temperature of 22 ± 2 °C with a relative humidity of 55% ± 5.2. After fasting for 18 h before the test, the mice were divided into 4 homogeneous batches of 3 mice according to weight. The diameter (Do) of the right hind leg of each mouse was measured one hour before the different treatments using an electronic caliper. The treatment was done orally (gavage) as follows:Group 1: animals served as controls and received only physiological water at 0.2 mL per 25 g body weight;Group 2: animals received amygdalin extract of the MJ variety at 1000 mg/kg body weight;Group 3: animals received amygdalin extract of the BEN variety at 1000 mg/kg body weight;Group 4: animals were used as a reference and received a reference anti-inflammatory drug, salicylic acetic acid, at 50 mg/kg body weight.

One hour after gavage, 0.1 mL of a 5% formalin solution was injected into each mouse under the plantar pad of the right hind paw. The paw diameter at the arch was measured every hour until the fifth hour using electronic display calipers. Edema increase (AOP) and inhibition (IOP) percentages were calculated according to the formula:$$\mathrm{PAO}=\frac{\mathrm{Dt}-\mathrm{Do}}{\mathrm{Dt}}\times 100$$
where Dt: mean diameter of the right hind leg at time t; Do: mean diameter of the right back leg at time 0 (before treatment).

### 4.10. Evaluation of Anticancer Activity

This was performed in vivo with Wistar rats following the protocol adapted from [68]. All handling procedures also considered the guidelines for studying animal models of cancer. The effect of amygdalin extracted from the BEN variety (the variety with the best antibacterial and anti-inflammatory activities) of cassava on unchecked cell proliferation in rats was tested by analysis of biochemical parameters and histopathological observations. 1,2-dimethylhydrazine (DMH) was used to induce colon cancer in vivo. It was prepared in 1 Mm EDTA saline at pH 7.1 for subcutaneous injection of 45 mg/kg of body weight. The reference molecule was 5-fluorouracil at 0.6 mL/kg/body weight. The extract was administered by gavage at 200 mg/mL in a volume of 100 mL/kg body weight.

#### 4.10.1. Experimental Design

Rats were randomly matched and body-marked for indication. The 18 rats were divided into six groups of 3 according to their body weight. All animals were provided food and water ad libitum throughout the experiment.

Group 0: negative control received plain water and food during the six weeks of the experiment.Group 1: inhibitory effect of the extract. These rats received a twice-weekly subcutaneous injection of DMH plus daily gavage of the amygdalin extract.Group 2: positive control, treated with DMH. Rats received a subcutaneous injection of DMH twice weekly for the five weeks of the experiment.Group 3: curative effect of extracts. Rats were injected subcutaneously with DMH twice a week for two weeks, and at the third week, a daily extract dose was applied until the end of the five weeks.Group 4: preventive effect of extracts. Rats were given amygdalin extract daily for three weeks, and then from the fourth week onwards, subcutaneous injections of DMH twice a week for the next two weeks.Group 5: effects of the reference molecule. Rats received a subcutaneous injection of DMH twice a week, combined with an intravenous infusion of 5-fluorouracil.

1,2-dimethylhydrazine (DMH) was prepared fresh weekly in EDTA saline with a pH adjusted to 7.1 using NaOH solution immediately before subcutaneous injection at a dose of 45 mg/kg body weight DMH. The follow-up time for the rats was six weeks for all batches.

#### 4.10.2. Determination of Biochemical Parameters

Blood samples were taken by puncturing the retroorbital sinus of the animals (under diethyl ether anesthesia). In addition to the weight of the animals, biochemical parameters (hematocrit, hemoglobin, red blood cells, white blood cells, leukocytes, and neutrophils) were measured before and at the end of the experiment from the animals’ whole blood samples. In addition, plasma creatinine (REF 80107, LOT 012042A) and urea (REF 92032, LOT 041915A) concentrations, as well as aspartate aminotransferase (AST) and alanine aminotransferase (ALT) (REF 51830, LOT 072105A7), were determined with BIOLABO (FRANCE) diagnostic kits on plasma collected after centrifugation of blood at 3000 rpm.

#### 4.10.3. Anatomical Observations

After six weeks of follow-up, all animals were sacrificed under the same fasting conditions, in the evening (22 h), under yellow light, according to the protocol described by [68]. The liver, kidneys, and colon were removed and flushed with saline. The colon was opened longitudinally and examined microscopically to observe the presence of multiple identifiable lesions and the appearance of aberrant crypt foci (ACF). Colon segments containing multiple plaque lesions (MPL) were dissected, fixed immediately in 10% formalin over kerosene, sectioned, and stained with hematoxylin and eosin for histopathologic observations. The liver and kidneys were placed in 10% formalin for histological sections.

#### 4.10.4. Statistical Analysis

GraphPad Prism 8 software was used for the study of variances. The Newman–Keuls test (SNK) was applied in the first step to compare the average amygdalin contents of the tested organs and derivatives. In the second step, the Student’s *t*-test was used to compare the moderate amygdalin content within each organ.

## 5. Conclusions

The present study evaluated the antioxidant, anti-inflammatory, and anticancer activities of amygdalin extracted from organs and derivatives of Benin’s most produced cassava varieties. The survey carried out on the powders obtained from the leaves of *Manihot esculenta* allowed us to identify some large families of its molecules, among which we can quote alkaloids, flavonoids, tannins, glycosides, saponosides, steroids, leuco-anthocyanins, coumarins, etc.

Cytotoxicity tests did not reveal any toxic effects of the extracts at the dose studied. So, this work presents the reasons that underlie the results of our investigations, stipulating the strong production and transformations of the three varieties of cassava (BEN, RB, and MJ) in Benin.

This study highlights all the organs (leaves, stems, roots, peelings, and second skin) of the three varieties of *Manihot esculenta*, some of which were neglected by the population, thus causing environmental pollution. It also enhances the value of the by-products resulting from cassava processing in Benin. This can contribute to the fight against problems related to food insecurity and the improvement of living conditions for our populations, on the one hand, and to the prevention and relief of chronic non-communicable diseases, on the other.

The study evaluated the in vitro and in vivo biological activities of amygdalin extracts from the plant. The results revealed that cassava varieties grown in Benin possess remarkable antioxidant activities and anti-inflammatory properties similar to those of non-steroidal anti-inflammatory drugs (NSAIDs). In addition, amygdalin extracts from BEN cassava leaves had anticancer activities in cancer-prone Wistar rats. This molecule effectively prevents and cures cancer by inhibiting the proliferation of cancer cells induced by DMH, which is confirmed by biochemical analysis of blood samples and histological examination.

Indeed, amygdalin, a molecule known for its numerous pharmacological properties, was quantified in cassava organs and derivatives. The results revealed the presence of this molecule in all samples but at variable levels. The leaves of the different cassava varieties and the *Agbeli* derivative showed a higher level of amygdalin. The anticancer power of the plant is due to the presence of a high level of amygdalin in its leaves.

## Figures and Tables

**Figure 1 molecules-28-04548-f001:**
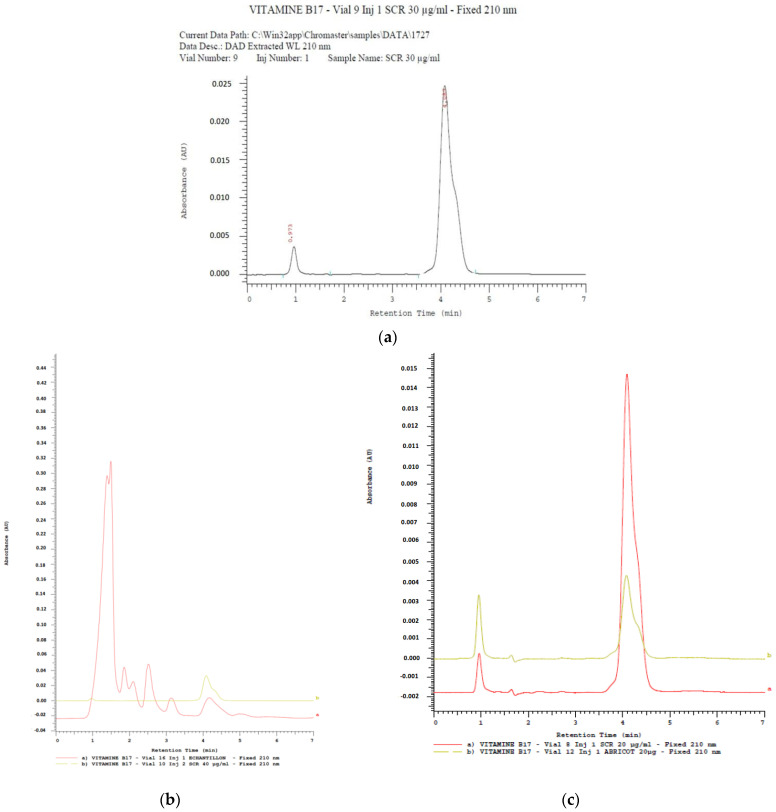
Chromatogram of amygdalin standard (**a**); standard sample (**b**) and standard derived (**c**).

**Figure 2 molecules-28-04548-f002:**
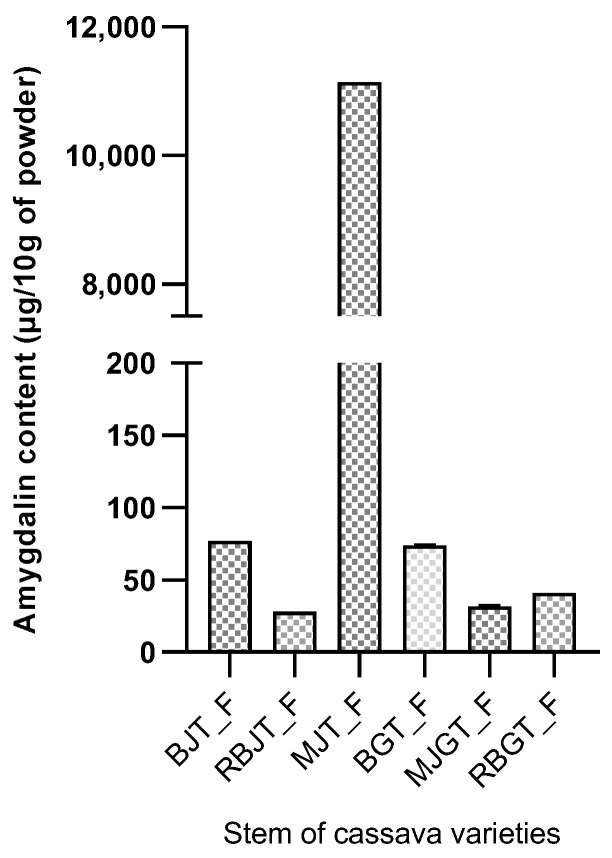
Amygdalin content (µg/10 g of cassava stem powder). Legend: BJT_F = BEN Young Fresh Stem; RBJT_F = RB Young Fresh Stem; MJT_F = MJ Young Fresh Stem; BGT_F = BEN Large Fresh Stem; MJGT_F = MJ Large Fresh Stem; RBGT_F = RB Large Fresh Stem.

**Figure 3 molecules-28-04548-f003:**
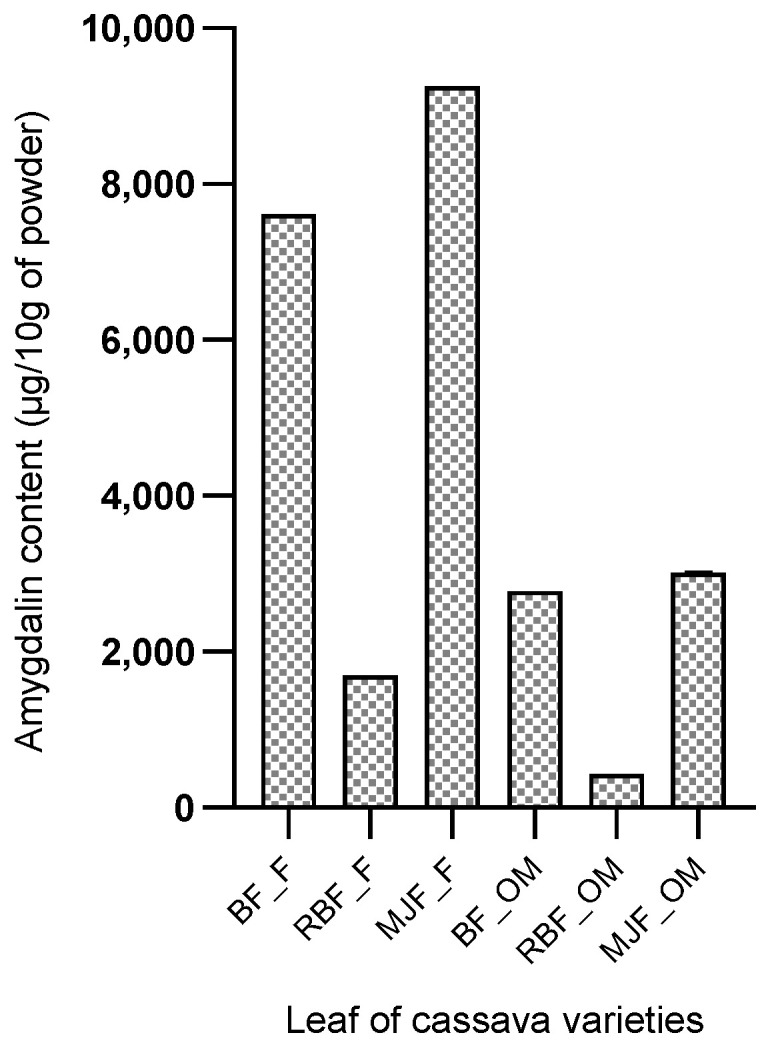
Amygdalin content (µg/10 g of cassava leaf powder). Legend: BF_F = BEN Fresh leaf; RBF_F = RB Fresh leaf; MJF_F = MJ Fresh leaf; BF_OM = BEN Shade dried leaf; RBF_OM = RB Shade dried leaf; MJF_OM = MJ Shade dried leaf.

**Figure 4 molecules-28-04548-f004:**
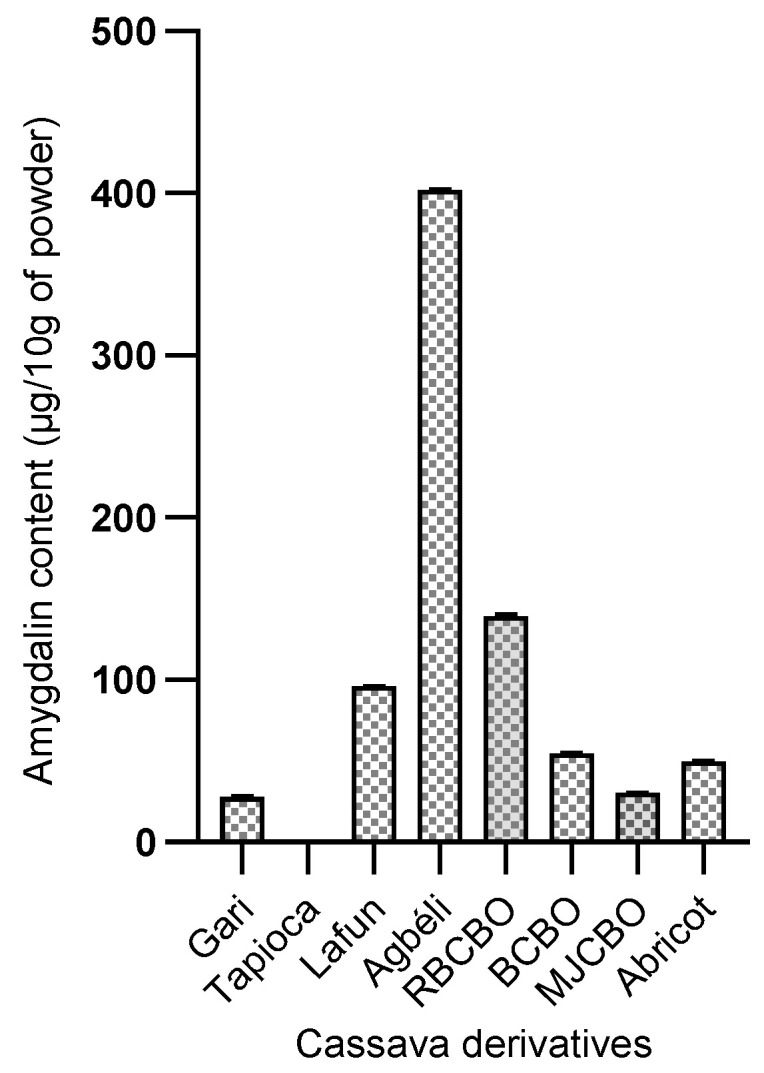
Amygdalin content (µg/10 g of cassava derivatives powder). Legend: RBCBO = RB Boiled Flesh; BCBO = BEN Boiled Flesh; MJCBO = MJ Boiled Flesh.

**Figure 5 molecules-28-04548-f005:**
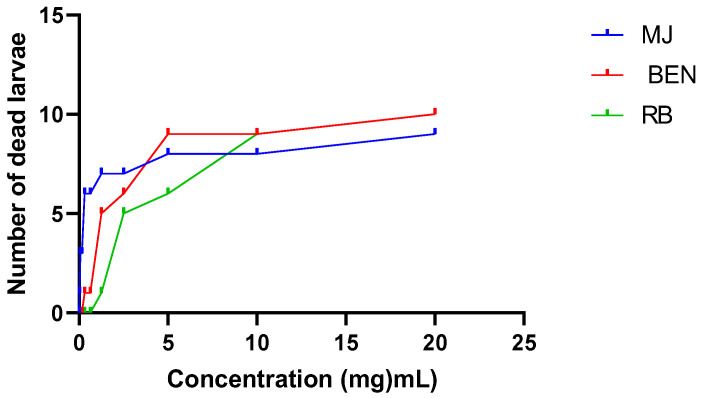
Larval cytotoxicity of extracts.

**Figure 6 molecules-28-04548-f006:**
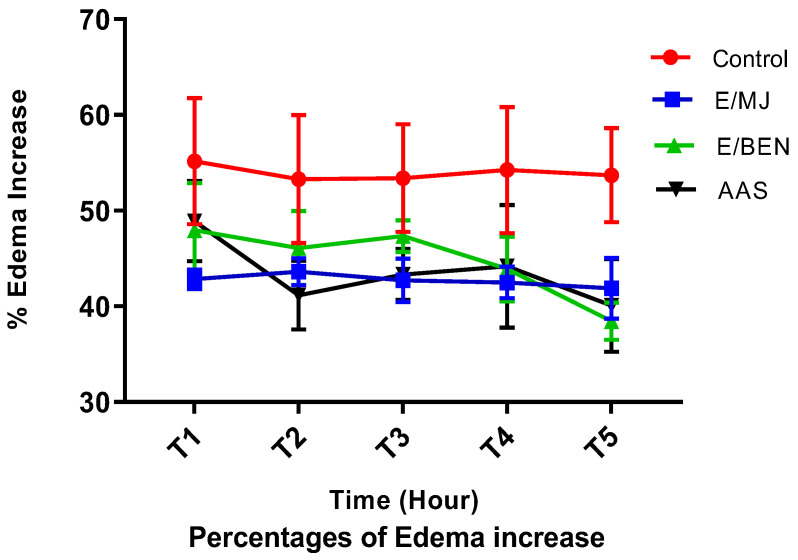
Percentage increase in hind paw edema in mice as a function of time. Legend: E/MJ (amygdalin extract of MJ variety); E/BEN (amygdalin extract of BEN variety); AAS (acetyl salicylic acid).

**Figure 7 molecules-28-04548-f007:**
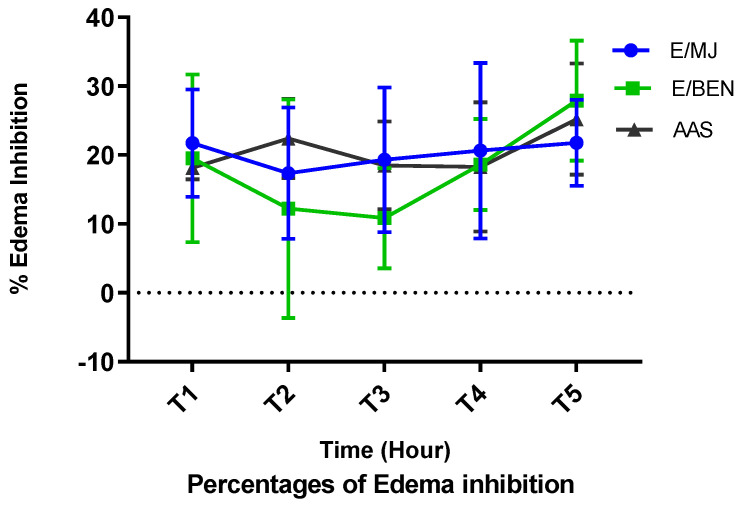
Percentage inhibition of hind paw edema in mice as a function of time. Legend: E/MJ (amygdalin extract of MJ variety); E/BEN (amygdalin extract of BEN variety); AAS (acetyl salicylic acid).

**Figure 8 molecules-28-04548-f008:**
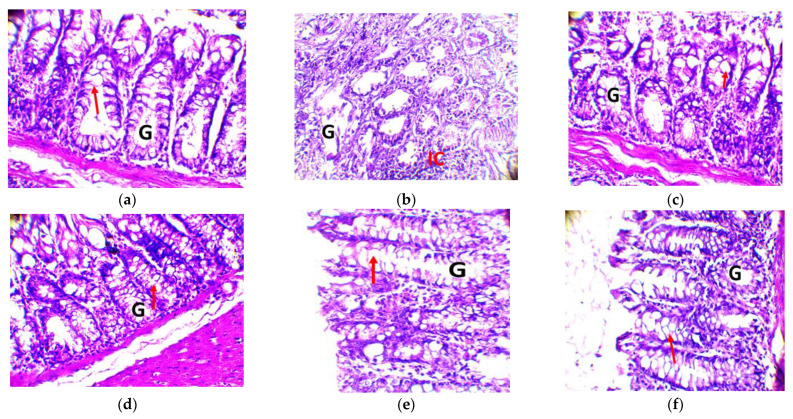
Histology of experienced Wistar rat colon (original magnification 400×). Legend: Lieberkühn gland (G); cell infiltrates (IC; caliciform cells (arrows). Magnification: 4000×. (**a**) group 1: DMH and amygdalin; (**b**) group 2: DMH only; (**c**) group 3: DMH, then amygdalin next week; (**d**) group 4: amygdalin, then DMH next week; (**e**) group 5: DMH and 5-fluorouracil combined; (**f**) control group: no treatment.

**Figure 9 molecules-28-04548-f009:**
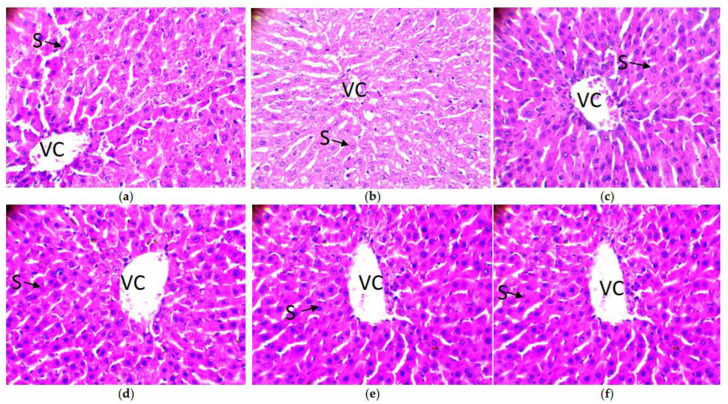
Histology of the liver of tested Wistar rats (original magnification 400×). Legend: venous sinusoids (S); centrilobular veins (VC); (**a**) group 1: DMH and amygdalin; (**b**) group 2: DMH only; (**c**) group 3: DMH, then amygdalin next week; (**d**) group 4: amygdalin, then DMH next week; (**e**) group 5: DMH and 5-fluorouracil combined; (**f**) control group: no treatment. Black arrow show venous sinusoids.

**Figure 10 molecules-28-04548-f010:**
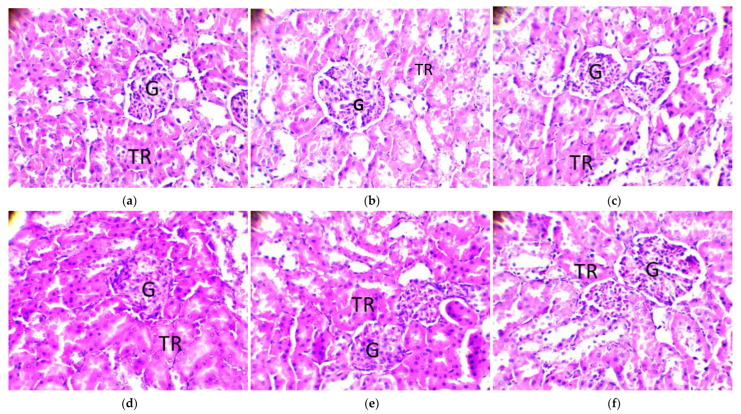
Histology of the kidneys of tested Wistar rats (original magnification 400×). Legend: glomeruli (G); renal tubules (RT); (**a**) group 1: DMH and amygdalin; (**b**) group 2: DMH only; (**c**) group 3: DMH, then amygdalin next week; (**d**) group 4: amygdalin, then DMH next week; (**e**) group 5: DMH and 5-fluorouracil combined; (**f**) control group: no treatment.

**Table 1 molecules-28-04548-t001:** Result of the phytochemical screening test.

Secondary Metabolites	Cassava Varieties Leave		
	BEN	MJ	RB
Alkaloids	+	+	+
Tannins	+	+	+
Saponosides	+	+	+
Leuco-anthocyanins	+	+	+
Flavonoids	+	+	+
Steroids	+	+	+
Triterpenes	-	-	-
Coumarins	+	+	+
Glycosides	+	+	+
Cyanogenic derivatives	+	+	+

Legend: (+): presence of metabolite; (-): absence of metabolite.

**Table 2 molecules-28-04548-t002:** Yield of extracts from sun- and shade-dried samples.

Samples	Organs	Yield (%)	
		Drying in the Sun	Drying in the Shade
MJ	Chair	11.40	9.23
	1st skin	1.36	1.18
	2nd skin	14.20	14.78
	Leaves	-	1.50
BEN	Chair	12.41	3.55
	1st skin	1.13	1.05
	2nd skin	12.26	10.95
	Leaves	-	1.5
RB	Chair	13.08	4.65
	1st skin	0.98	0.95
	2nd skin	18.21	12.75
	Leaves	-	1.50

**Table 3 molecules-28-04548-t003:** Total flavonoid, polyphenols, and tannin content of ethanolic leaf extracts of cassava varieties (mean ± standard deviation).

Cassava Varieties	Flavonoids (μgEQ/100 mg)	Polyphenols (μgEAG/100 mg)	Tannins (mgEAG/g Extract)
BEN	110.96 ± 1.18 ^a^	52.59 ± 7.56 ^b^	0.35 ± 0.07 ^b^
MJ	129.36 ± 9.22 ^a^	32.62 ± 8.70 ^c^	0.37 ± 0.04 ^b^
RB	125.20 ± 2.77 ^a^	65.14 ± 4.74 ^a^	0.54 ± 0.03 ^a^

Different lowercase letters (a, b and c) between columns indicate a significant difference (*p* < 0.05).

**Table 4 molecules-28-04548-t004:** In vitro antioxidant activity of amygdalin extracted from different organs of sun-dried cassava varieties by DPPH essay.

Substances	Extracts of Sun-Dried Organs	PI (%)	IC_50_ (µg/mL^−1)^
RB	Flesh	87.28 ± 1.25	3.00 ± 0.00
	1st skin	91.72 ± 0.00	0.75 ± 0.07
	2nd skin	91.15 ± 0.14	2.35 ± 1.20
MJ	Flesh	95.11 ± 0.18	0.5 ± 0.22
	1st skin	93.77 ± 0.54	0.25 ± 0.07
	2nd skin	85.12 ± 0.67	4.6 ± 1.97
BEN	Flesh	94.71 ± 0.07	<0.19
	1st skin	95.38 ± 0.07	<0.19
	2nd skin	89.38 ± 0.23	7.25 ± 0.35

**Table 5 molecules-28-04548-t005:** In vitro antioxidant activity of amygdalin extracted from different organs of shade-dried cassava varieties.

Substances	Extracts of Organs Dried in the Shade	PI (%)	IC_50_ (µg/mL^−1^)
Reference molecule	Quercetine	82.35 ± 1.86	5.75 ± 1.06
RB	Flesh	88.93 ± 0.12	17.25 ± 0.35
	2nd skin	89.60 ± 0.98	4.1 ± 0.14
MJ	Flesh	90.14 ± 0.32	21 ± 14.14
	2nd skin	92.10 ± 0.16	0.5 ± 0
BEN	Flesh	85.88 ± 0.28	9.6 ± 0.56
	2nd skin	91.37 ± 0.18	2.75 ± 0.35

**Table 6 molecules-28-04548-t006:** Antioxidant activity (IC_50_ µg. mL^−1^) of cassava varieties leaves by DPPH and FRAP essays.

Standard/Samples	IC_50_ (µg mL^−1^)	
	DPPH	FRAP
Ascorbic acid	1.11 ± 0.09	-
RB	8.11 ± 0.70	0.63 ± 0.04
MJ	3.25 ± 0.32	0.69 ± 0.03
BEN	3.99 ± 0.27	0.52 ± 0.04

**Table 7 molecules-28-04548-t007:** LC_50_ of ethanolic extracts from cassava varieties leaves.

Amygdalin Extracts	RB	BEN	MJ
LC_50_ (mg/mL)	12.73 ± 0.07	11.60 ± 0.96	12.32 ± 0.17
R^2^	0.80 ± 0.05	0.64 ± 0.75	0.41 ± 0.09

**Table 8 molecules-28-04548-t008:** Variations in biochemical parameters of rats in each batch.

Variables	R0	R1	R2	R3	R4	R5	*p*-Value	Significativity
Hb	13.66 ^b^ ± 0.49	16.43 ^a^ ± 0.9	9.00 ^c^ ± 1.0	15.6 ^a^ ± 0.70	14.2 ^b^ ± 0.30	13.6 ^b^ ± 0.60	7.215 × 10^−7^	***
Hte	41.00 ^b^ ± 2.00	9.00 ^c^ ± 1.00	19.33 ^d^ ± 2.00	30.33 ^c^ ± 3.20	42.00 ^b^ ± 2.00	39.33 ^b^ ± 1.50	6934 × 10^−8^	***
NR	4.43 ^b^ ± 0.26	5.46 ^b^ ± 0.30	10.79 ^a^ ± 2.50	6.16 ^b^ ± 1.80	7.62 ^b^ ± 2.40	4.26 ^b^ ± 0.00	0.003153	**
NB	7.77 ^ab^ ± 1.15	4.18 ^b^ ± 0.80	13.82 ^a^ ± 6.10	7.91 ^ab^ ± 2.40	8.86 ^ab^ ± 0.40	7.12 ^ab^ ± 1.00	0.02841	*
Lym	28.00 ^c^ ± 7.21	39.33 ^c^ ± 7.20	114.66 ^a^ ± 7.30	84.00 ^b^ ± 11.10	37.33 ^c^ ± 11.90	21.66 ^c^ ± 1.50	5.907 × 10^−8^	***
Neut	67.33 ^b^ ± 11.84	61.66 ^b^ ± 6.40	114.33 ^a^ ± 8.50	82.00 ^b^ ± 16.30	62.33 ^b^ ± 11.50	72.00 ^b^ ± 11.20	0.0008887	***
Eos	0.33 ^b^ ± 0.57	0.33 ^b^ ± 0.50	1.66 ^a^ ± 0.50	00 ^b^ ± 00	0.33 ^b^ ± 0.50	00 ^b^ ± 00	0.009067	**
Mono	0.33 ^b^ ± 0.57	00.00 ^a^ ± 00.00	00.00 ^a^ ± 00.00	0.66 ^a^ ± 1.10	00.00 ^a^ ± 00.00	0.66 ^a^ ± 0.50	0.4755	NS
ASAT	97.66 ^b^ ± 9.07	96.66 ^b^ ± 4.04	570.33 ^a^ ± 39.80	147.33 ^b^ ± 46.50	121.00 ^b^ ± 8.80	102.33 ^b^ ± 2.50	1.427 × 10^−10^	***
ALAT	86.33 ^c^ ± 10.21	67.33 ^c^ ± 10.10	567.66 ^a^ ± 55.01	160.33 ^b^ ± 26.50	122.66 ^bc^ ± 8.10	91.00 ^c^ ± 5.20	1.311 × 10^−10^	***
CREAT	6.92 ^c^ ± 0.56	8.10 ^c^ ± 0.40	19.57 ^a^ ± 1.50	8.21 ^c^ ± 0.80	13.10 ^b^ ± 5.30	7.60 ^c^ ± 1.40	0.0001756	***

R1: amygdalin and DMH simultaneously (inhibition effect of the extract); R2: treated with DMH; R3: DMH then amygdalin (inhibition effect of the extract); R4: amygdalin, then DMH (preventive effect of the extract); R5: effects of 5-fluorouracil used as reference molecule. HB: hemoglobin level; Hte: hematocrit level; NR: red cell; NB: white cell; Lym: lymphocyte; Neut: neutrophils; Eos: eosinophil; Mono: monocyte; ASAT: aspartate aminotransferase; ALAT: alanine aminotransferase; CREAT: creatinine. Different lowercase letters (a, b and c) between rows indicate a significant difference (*p* < 0.05). NS: not significant difference (*p* > 0.05); * *p* < 0.05; ** *p* < 0.01; *** *p* < 0.001.

**Table 9 molecules-28-04548-t009:** Characteristics of cassava varieties and their derived products.

Name of the Varieties	Characteristics	Quality and Different Processing Options
	Soft variety	
	Brown stem	
BEN 86052	Dark green leaf, edible.	Gari, Tapioca, Agbéli, lafoun
	Average yield in pods: 30%.	
	Soft variety	
	Brown stem	
RB 89509	Green leaf, edible	Gari, Tapioca, Agbéli, lafoun
	Low hydrocyanic acid content	
	Soft variety	
	White stem	
Yellow cassava (MJ)	Edible leaf	Gari, semolina
	Low hydrocyanic acid content	

## Data Availability

Not applicable.

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
