# Peer review of "Antioxidant, Anti-Inflammatory, and Anti-Cancer Properties of Amygdalin Extracted from Three Cassava Varieties Cultivated in Benin"

_molecules, 2023, doi:10.3390/molecules28114548_

Round 1

Reviewer 1 Report

The manuscript titled "Antioxidant, anti-inflammatory and anti-cancer properties of amygdalin extracted from three cassava varieties cultivated in Benin" investigates the potential medicinal properties of cassava plants, which are widely grown in Benin. Specifically, the study examines the biological activities of amygdalin, a natural compound found in cassava, and evaluates its potential as an antioxidant, anti-inflammatory, and anticancer agent.

The authors of the study used HPLC analysis to measure amygdalin content in cassava organs and derivatives, performed phytochemical screening to identify secondary metabolite groups, and conducted in vivo tests to evaluate the compound's various properties. The results of the study showed that all three cassava varieties studied contained glycosides, flavonoids, saponosides, steroids, tannins, coumarins, and cyanogenic derivatives. Additionally, the leaves of the cassava plants contained high levels of amygdalin, and extracts from these leaves showed potential as anti-inflammatory and anticancer agents.

Overall, the study provides valuable insights into the medicinal properties of cassava and specifically, the potential of amygdalin as a natural compound for treating cancer and inflammation.

Major issues identified include:

1) It is important for the manuscript to address the potential toxicity of amygdalin, given that cyanogenic glycosides are natural plant toxicants. The action of endogenous plant enzymes can release hydrogen cyanide, which may pose a risk of toxicity for animals, including humans. It would be beneficial to have a discussion on this topic in the introduction section to demonstrate the authors' awareness of the potential toxicity of amygdalin, in addition to its potential health benefits. As the saying goes, "every drug is a poison; it's a matter of dose."

2) The reported presence of amygdalin (vitamin B17) in cassava plants is based on the comparison of HPLC-UV chromatograms of extracts and a standard. However, the observed peak appears to be very broad, suggesting the possibility of the presence of other structurally similar compounds. Therefore, the HPLC method should be optimized to achieve better resolution of the amygdalin peak. It is evident that the peak does not have a Gaussian shape, indicating the presence of more than one compound that needs to be resolved.

3) In the chromatograms in Figure 1, wavelength was set to 210 nm. However, the HPLC analysis (section 4.6.2) reports the wavelength of 280 nm. Please explain this discrepancy. Since the detection was only UV-Vis, the wavelength of detector will have significant influence on the observed HPLC profile. In addition, section 4.6.2. contains a lot of unnecessary data (type of monitor, mouse etc.)

4) Pay attention to the number of significant figures in Figures 2-4. Round the results to 2 decimal places.

5) How did the authors determine the extraction yield (2.2) and what compound(s) was extracted?

I have identified several issues with the manuscript that need to be addressed. Firstly, the language, flow, and clarity of this version of the manuscript are poor. The authors need to improve the writing style and organization of the manuscript to make it more readable and understandable.

Reviewer 2 Report

Dear Author,

Kindly response to the below comments:

Figure 1. Chromatogram of amygdalin standard [a]; standard-sample [b] and standard-derived

[c]: what do you mean by standard-sample and standard-derived? Kindly give more details

Figure 2, 3 and 4: kindly use mg and keep only 2 decimals

Figure 2: please give an explanation why MJT_F is very rich in Amygdalin but all the other sample almost there is no Amygdalin?

Figure 5: there is no control for the cytotoxicity test?

Figure 6: which is the used control?

Figure 8 e: please change this photo

Kindly added more details regarding the cassava varieties and derivatives

Minor editing of English language required

Reviewer 3 Report

The manuscript entitled "Antioxidant, anti-inflammatory and anti-cancer properties of amygdalin extracted from three cassava varieties cultivated in Benin" is a good work by the authors. The work was designed well and methodology was technically sound. However, I recommend some changes prior to further consideration.

1. 11142.99 µg can be converted to higher concentration like mg and it can be expressed per Unit weight (per gram) 

2. It is also recommended to included standard deviation values in all the experimental results

3. The values in table 2 needs to be in Mean and SD format as mentioned above

4. The figure 2 must be modified (especially MJT_F) by splitting Y xis to show other bars clearly

5. In all figures, standard deviation must be indicated

6. Plant collection location must be indicated and authentication must be given

7. What was the reason for choosing 3 rats per group? How the minimum number of animals per group was determined? Is it sufficient to analyze the results? Necessary explanation and literature support must be provided

There are some punctuation and spelling errors in the manuscript. It needs to be addressed.

Round 2

Reviewer 1 Report

The authors have addressed my comments and corrected the manuscript accordingly.

The quality of English Language is acceptable.

Reviewer 3 Report

No more comments.